



# Relationship between the stocks of carbon in non-cultivated trees and soils in a West-African forest-savanna transition zone

Tegawende Léa Jeanne Ilboudo[1, 6*] Lucien NGuessan Diby[2], Delwendé Innocent Kiba[3, 6], Tor Gunnar Vågen[4], Leigh Ann Winowiecki[4], Hassan Bismarck Nacro[1], Johan Six[5], Emmanuel Frossard[6]

[1]Sciences Naturelles et Agronomie, Université Nazi Boni, Bobo Dioulasso, 01 BP 1091 Bobo-Dioulasso 01, Burkina Faso
[2]Institut National Polytechnique Félix HOUPHOUËT-BOIGNY (INP-HB), Ecole Supérieure d'Agronomie (ESA), BP 1313 Yamoussoukro, Côte d'Ivoire
[3]Centre National de la Recherche Scientifique et Technologique (CNRST), Institut de l'Environnement et de Recherches Agricoles (INERA), Laboratoire Ressources Naturelles et Innovations Agricoles (LARENIA), 01BP 476 Ouagadougou 01, Burkina Faso [4] World Agroforestry Centre (ICRAF), P.O. Box 30677-00100, Nairobi, Kenya
[5]Group of Sustainable Agroecosystems, Swiss Federal Institute of Technology, Zurich, 8092 Zurich, Switzerland
[6]Group of Plant Nutrition, Swiss Federal Institute of Technology, Zurich, 8315 Lindau, Switzerland

*Correspondence to*: T. L. Jeanne Ilboudo (tilboudo20@gmail.com)

**Abstract.**

Carbon is a key element for the functioning and productivity of tropical soils. While the impact of organic inputs on carbon storage in these soils is known, little is known about the contribution of non-cultivated trees.

In this study, we measured carbon content in non-cultivated trees (VC), soil organic carbon (SOC) and soil total nitrogen (TN) in different land uses in a West African forest – savanna transition zone. We used the Land Degradation Surveillance Framework for data collection and allometric equations to estimate the stocks of VC on a 10 km * 10 km landscape. Soil samples were taken in 160 sites at 0 – 20 cm, 20 – 50 cm, 50 – 80 cm and 80 – 110 cm depth in different land uses. We developed Partial Least Square regression models to predict SOC, TN and clay concentrations from mid-infrared soil spectra. We then considered soil bulk density to calculate the stocks of SOC and TN for each sampling depth and conducted a path analysis to identify the factors controlling these parameters. Our results showed that at landscape level, tree density and diversity explained most of VC stocks variability. SOC stock variability was mainly explained by clay content. The main drivers of TN stocks were clay and SOC stock. The VC and SOC stocks were not correlated with each other when considering all data. However, we found significant linear positive relationships between VC and SOC stocks for the land uses annual croplands, perennial croplands, grasslands and bushlands without soil depth restrictions until 110 cm. We concluded that in the forest-savanna transition zone, soil properties and topography determine land use, which in turn affects the stocks of SOC and TN and to some extent the VC stocks. Bushlands conservation and perennial trees cropping systems could be recommended for improved SOC storage.

*Key words: Carbon and nitrogen stocks, land use, forest – savanna transition and West Africa*



## 1 Introduction

Organic carbon is a key factor for soil functioning (Hagedorn et al., 2018). It is also a primary sink for greenhouse gases and
therefore has a direct impact on climate change (Sreenivas et al., 2016; Wang et al., 2017). Carbon sequestration in soil or in vegetation mitigates climate change by reducing the $CO_2$ concentration in the atmosphere (Lal, 2006, Reyns, 2020). Soil organic carbon (SOC) plays a key role in agricultural production particularly in the highly weathered tropical soils because organic matter is both a sink and source of plant nutrients due to the low cation exchange capacity of the most abundant clay mineral kaolinite in these soils (Khawmee et al., 2013).

The stocks of organic carbon in soils result from inputs by the vegetation, organic fertilizers, outputs through mineralisation, erosion, and stabilisation in the soil matrix through interactions with clay minerals and metallic oxi-hydroxides. These stocks vary with vegetation, precipitation, parent material, soil texture, land use and topography (Saiz et al., 2012; Zhu et al., 2019) and are determined by any factor that could impact net primary productivity and soil respiration (Davidson and Jansens 2006). The inputs of carbon from vegetation depend on many factors as the type, density and the diversity of this vegetation. For
instance, strong positive correlations between the stock of carbon in trees in one hand and tree density, size and height on the other hand were highlighted (Aryal et al., 2018; Dimobe et al., 2019). The impact of the vegetation diversity on its carbon storage were however contrasted. Indeed, Li et al. (2019) showed a positive effect of tree diversity on tree carbon stock while some authors highlighted the opposite (Henry et al., 2009; Aryal et al., 2018). Deng et al., (2013) showed a gradual increase of soil organic carbon stocks following a succession going from grassland to shrubland to woodland.

In cropland soils, organic fertilization inputs may increase the total carbon input by 30-80% according to Xia et al. (2017). However, in some cropping systems as those from the African smallholder farms, these exogenous organic inputs are limited due to poor integration between crops and livestock and also due to limited labor for manure processing and application (Mutyasira et al., 2018). Conservation agriculture because of the minimum soil disturbance contribute to a reduced erosion and mineralization of SOC (Debaeke et al., 2017) and therefore to an increased carbon storage in soils (Guo et al., 2016).
However, this cropping practice is limited by the competition with livestock for crop residues and the difficulty of controlling weeds without herbicide application (Debaeke et al., 2017).

Soil organic carbon losses are mainly related to mineralization, leaching, and erosion (Gabarrón-Galeote, et al., 2015) and thus strongly depend on soil microbial composition and climatic conditions. In the tropics, the losses of SOC can result from a variety of factors. The use of wood as the main source of energy leads to deforestation, followed by low carbon sequestration
and increased soil losses through erosion (Pearson et al., 2017). In addition, inappropriate cropping practices contribute significantly to increased losses of SOC. For example, slash and burn and conventional tillage lead to a decrease in the stocks of SOC (Kukla et al., 2019; Liu et al. 2016.). The high temperatures in the tropical regions also accelerate the mineralization and thus lead to some increased losses of SOC from the ecosystems (Bashan and de-Bashan, 2010).

Whereas a lot of information is available on SOC stocks in temperate ecosystems, there is comparatively less information for
tropical soils from Africa. Menezes et al. (2021) state, for example, that the lack of scientific information prevents accurate



estimation of soil and plant carbon stocks, mainly in tropical regions, while significant losses of forest area are occurring in these regions.

A characteristic of African ecosystems has been the recent changes in land use from little disturbed forests or savanna to cropped lands. In sub-Saharan Africa, for example, increased conversion of natural land to arable land has been caused by
rapid population growth and economic development (Kleeman et al., 2017; Yira et al 2016). In African croplands, farmers manage non-planted trees in agroforestry systems that provide several socio-economic and biophysical ecosystem services. Since organic compounds can be stabilized over decades in soils, the organic matter present in those croplands should rather reflect the inputs of the vegetation present before soils started to be cropped as these were dominated by non-cultivated trees. In this study we hypothesized that the relationship between the stock of soil organic C and the stock of C in the uncultivated
will not be identical in all land uses. For instance, in land uses that are not cropped we should observe a positive relation between these two types of stocks as uncultivated trees are probably still an important source of C for the soil, whereas in land uses managed with annual crops, from which a lot of uncultivated trees have been removed and where soils are annually disturbed, we should observe no relation between C stocks in uncultivated trees and in soils. We tested this hypothesis by analysing the stocks of C in the non-cultivated trees and in the soils of a forest-savanna transition zone in the centre of Côte
d'Ivoire where more or less intensively managed land uses were to be found.

## 2 Materials and methods

### 2.1 Study area

The study was conducted in Tiéningboué (5.72° W; 8.18° N), located in the centre of Côte d'Ivoire, West Africa (figure 1). This area is classified as a tropical savanna according to Koppen-Geiger climate classification (Peel et al., 2007). From 2005
to 2014, the mean annual total precipitation of Tiéningboué was $1241 \pm 146$ mm (NASA, 2021), data at 25 km resolution. The mean temperature was $29°C \pm 2.5°C$ based on data from the Moderate-Resolution Imaging Spectroradiometer (USGS, 2021) at 5.6 km resolution. The soils are derived from granitic and granito-gneiss rocks (Camara, 1983; Diatta, 1996) leading to the development of Plinthosols with accumulation of iron oxides at depth (Jones et al., 2013). Other soil groups such as Acrisols, Luvisols and Lixisols are also encountered (FAO, 2021).

### 2.2 Data collection

We collected the data in June 2015 using the Land Degradation Surveillance Framework (LDSF) (Vågen et al. 2010). The LDSF uses a standardized hierarchical field sampling design to provide a biophysical baseline at landscape level and a monitoring and evaluation framework to assess processes of land degradation (Vågen and Winowiecki, 2013). According to this method, we defined a sentinel site of 10 km x 10 km divided into 16 equal clusters. Each cluster contained a random
centroid with 10 sampling plots in which data were collected. The LDSF as it uses a nested hierarchical sampling design allows for the development of predictive models that has a global coverage without changing the local relevance.



Land use classification was done using a simplified version of the FAO Land Cover Classification System (LCCS) (Di Gregorio and Jansen, 1998). In addition, we separated annual croplands from perennial croplands due to their presumably different contribution to soil carbon stocks. For each of the LDSF plot, we collected information on the geographic coordinates and altitude, the slope and topography and on the impacts of erosion, fire, and grazing.

Non-cultivated tree is defined in our context as a living tree that was present in the sampling plot and was not planted by the producer as a main crop. In the plot, non-cultivated tree species were identified and the measurements of tree height was done using a Laser Ace 1000 and diameter at breast height using tape measure. Only non-cultivated trees with diameter $\geq 2.5$ cm and height $\geq 1.5$ m were considered in the annual croplands, we collected tree data within a 50 m radius while in the other plots, trees data were collected within 25 m radius. The nomenclature of Lebrun and Stork (1991) was used for trees species identification. Trees diversity was expressed by Shannon index (equation 1) calculated using the multivariate statistical program (MVSP) 3.1.

$$H' = -\sum_{i=1}^{s} Pi.lnPi \qquad\qquad (1)$$

Where $Pi = ni$ / N; S = number of species recorded, N = Total number of trees sampled, $ni$ = Number of trees of species.

Soil samples were taken in the centre of each plot at $0 - 20$ cm, $20 - 50$ cm, $50 - 80$ cm and $80 - 110$ cm of soil depth. Soil depth restrictions before 110 cm were registered. Each soil sample was kept in a plastic bag and sent to the laboratory. Soil samples were air-dried then sieved at 2 mm. The weights of soil fine and coarse fractions were recorded, and soil fine fraction was used for the different analysis.

## 2.3 Soil mid infrared spectroscopy

We used mid-infrared spectroscopy (MIRS) to analyse and predict soil total carbon, total nitrogen, and soil texture (Linker, 2008). Ten grams of air-dried soil fine fraction were milled at ~ 10 µm with MM 200 Retsch for spectral analysis using the Fourier Transform Infrared (FT- IR) spectrometer *Alpha* equipped with ZnSe optics and the scanning was done according to Terhoeven-Urselmans et al. (2010). The spectra were recorded from 4000 cm$^{-1}$ to 500 cm$^{-1}$ at 2 cm$^{-1}$ of resolution after 32 scans. Reference analysis was conducted on 15% of the soil samples, selecting samples using the Kennard-Stone algorithm (Kennard and Stone, 1969). Soil total carbon and total nitrogen contents in the reference soil samples were measured by combustion of 60 mg milled soil weighed into tin foil capsules (Vario PYRO cube, Elementar Analyse systeme GmBH). Given the acidic to neutral pH values of soils in West Africa, total carbon was assumed to be similar to soil organic carbon (SOC). The soil texture (2 mm) of the reference samples was also determined by the laser diffraction (Eshel et al., 2004) after mixing with calgon for four minutes (LA-950V2 Particle analyzer).

Packages in R software were used for spectra pre-processing and predictions. Spectra were pre-processed using plot function to visualize and identify bad spectra. Which function was used to exclude bad spectra and outliers, t function for first derivatives. With the calibrate function, we developed PLS models of total C, total N, clay, silt and sand content using both spectral and chemical analyses data of the reference samples. Round function was used to select the reference samples. The





models were then calibrated with the 70 % of the reference samples and validated with the remaining 30 %. C, N and clay
models performed well with R² > 0.8 and RMSE < 9 (figures S2, S3, S4 in supplementary material). These models were
subsequently used to predict C, N and clay for all the soil samples.

## 2.4 Carbon and nitrogen stocks calculations

Tree aboveground biomass was calculated according to Chave et al. (2005) for dry forests and bushlands and Brown et al.
(1989) for savanna vegetation (croplands, grasslands and wooded grasslands). Roots biomass was estimated as 24 % of the
aboveground biomass (Brown et al., 1989) while the carbon stock per tree was estimated as 50% of the total biomass of the
tree (Chave et al, 2005). In this study, the sum of the carbon stocks for all the non-cultivated trees in each plot was considered
as the tree carbon (VC) stock per plot.

Soil organic carbon (SOC) stock for a given depth was calculated according to equation 2. Soil total nitrogen (TN) stock was
also calculated with the same equation including soil total nitrogen concentration instead of soil carbon concentration. The
Soil bulk density is routinely measured in the LDSF method and cannot not give realistic information given the variability of
coarse particle contents of our study sites. We thus used the bulk density caculated in Hounkpatin et al. (2018) for soils from
the centre west of Burkina Faso to correct the soil coarse particle contents.

$$SOC\ stock = SOC * BD * Depth * (1 - frag) * 100 \tag{2}$$

where SOC stock = soil organic carbon stock (t C ha$^{-1}$), SOC = soil organic carbon concentration in soil fines fraction (< 2
mm) measured in the laboratory (g kg$^{-1}$), BD = soil bulk density (g cm$^{-3}$), Depth in cm, frag = % volume of soil coarse fragments
/100 to account for soil coarse fraction since SOC was measured on the fine fraction only. 100 is used to convert the unit to t
C ha$^{-1}$.

We calculated the sum of SOC, TN and VC stocks for all soil depths and all plots and land uses, then extrapolated it to get the
SOC total, TN and VC stocks in tons for 100 km$^2$.

## 2.5 Path analysis for carbon and nitrogen stocks drivers

We used path analysis using the R software to identify the factors controlling the variability of the SOC, VC, and TN stocks.
Confirmatory factor analysis (CFA) function was used in this path analysis. CFA allows for the assessment of fit between
observed data and a conceptualized model that specifies the hypothesized causal relations between factors and their observed
indicator variables (Mueller and Hancock, 2015). We used empirical knowledge and assumptions to construct a conceptual
model (figure S4 in supplementary material). High tree density, large tree diameter and high tree height were assumed to
increase VC stock. Tree diversity is assumed to have direct effect on VC stock by influencing the wood density (Zanne et al.,
2009). Tree diversity could also have an indirect effect on VC stock by affecting tree density, tree height and tree diameter
through allelopathy or modifying soil litter quality (Weil and Brady, 2017).





Soil clay content and coarse content were assumed to affect SOC and TN stocks given the strong link between clay mineral and organic matter storage (Alemayehu and Teshome, 2021). Slope and erosion were assumed to influence directly soil clay content and indirectly SOC stock and TN stock. Fire was assumed to increase the impact of erosion by destroying aboveground biomass including non-cultivated trees and litter on the soil.

Measured data were used to build a path model. Path standardized coefficients and their significance levels were calculated by
CFA approach. Only the significant relations ($p < 0.05$) were kept in the model and the goodness of the path model was tested using comparative fit index (CFI) and root mean square error of approximation (RMSEA). CFI values range from zero to one, with large values suggesting a good fit. RMSEA varies from zero to one, the smaller value indicating better model fit (Hu and Bentler, 1999).

**2.6 Statistical analysis**

All statistics were done with R software. Means, standard deviation and standard error were calculated using ANOVA function. Means were compared using Tukey test in lsmeans package, means with same the letter are not significantly different.

**3 Results**

**3.1 Land use, morphology and soil characteristics**

In our study area of 100 km², we observed six land uses during the sampling period (table 1). The dominant land uses were wooded grasslands found in 31 % of the plots, followed by perennial croplands (24.3 %) and annual croplands (23.6 %) whereas forests covered only 1.4 % of the plots. Perennial croplands were represented by cashew plantations and 73 % of them were between 8 to 15 years old, and only 6 % were older than 15 years. The youngest cashew plantations (2 – 4 years old) were intercropped with food crops like yam, cassava, maize, pepper, plantain, while older plantations were associated with
papaya and pineapple. The annual crops were mainly yam (16 % of annual croplands), cotton (13 % of annual croplands), groundnut (10 % of annual croplands) and rice (3 % of annual croplands). Annual crops were cultivated in slash-and-burn systems and yam and cotton were first in crop rotation. Groundnut and rice fields were grown in rotation following yam or cotton. Yam and groundnut were associated with cassava and maize.

The altitude of the plots varied between 208 m to 462 m. In this landscape, 1 % of the plots was located on ridges, 2 % on
bottomlands, 2 % on uplands, 11 % on foot-slopes and 84 % on mid-slopes. Around 70 % of wooded grasslands were located on mid-slopes and 15 % on foot-slopes; about 74 % of grasslands were on mid-slopes while 7 % and 15 % were on foot-slopes and bottomlands respectively. More than 90 % of bushlands, croplands and forests were located on mid-slopes, only 17 % of the plots were observed on foot-slopes, bottomlands, uplands. Wooded grasslands were on the steepest slope, whereas bushlands, croplands were located on areas with the lowest slope (table S3 in supplementary material). Fire and erosion impacts
were observed in at least 80 % of the wooded grasslands, grasslands, and annual croplands (table S3 in supplementary

 

material). Grazing and trees cutting impacts were also mostly observed in wooded grasslands, grasslands and lands prepared for annual crops cultivation.

Soil texture differed between land uses (table 2). Soil clay content in the 0-20 cm depth were high in forests ($354 \pm 19$ g kg⁻¹), perennial croplands ($352 \pm 88$ g kg⁻¹), annual croplands ($326 \pm 89$ g kg⁻¹) and bushlands ($318 \pm 51$ g kg⁻¹) while wooded

grasslands ($240 \pm 82$ g kg⁻¹) and grasslands ($245 \pm 73$ g kg⁻¹) had lower clay content. In general, clay content increased with soil depth in all land uses. Less than 25 % of croplands and grasslands presented depth restriction while 44 % of bushlands and 70 % of wooded grasslands presented soil depth restrictions before 110 cm.

### 3.2 Non-cultivated trees characteristics

In total, we recorded 136 species from 106 genera and 40 families. The dominant species were *Sterculia tragacantha* in forests,

*Uapaca togoensis* in wooded grasslands, *Terminalia scimperiana* in grasslands, *Parkia biglobosa* in perennial croplands. Bushlands and annual croplands presented the same dominant species *Margaritaria discoidea*. Tree diameter, height, tree density and diversity varied with land use (Table 1). The non-cultivated trees present in perennial croplands had the largest average diameter ($113 \pm 64$ cm) followed by those observed in annual croplands ($63 \pm 34$ cm) while the non-cultivated trees located in the wooded grasslands had the smallest diameter ($48 \pm 17$ cm). The tallest non-cultivated trees were observed in the

perennial croplands ($10.6 \pm 3$ m) while the shortest were found in the wooded grassland ($6.7 \pm 2$ cm). Tree height was not significantly different between the forests, bushlands, and annual croplands. Tree density was higher in bushlands ($452 \pm 210$ trees ha⁻¹) and wooded grasslands ($402 \pm 190$ trees ha⁻¹) than grasslands ($154 \pm 92$ trees ha⁻¹). Perennial croplands had the lowest non-cultivated tree density ($15 \pm 15$ trees ha⁻¹) and tree species diversity. Tree species were more diverse in bushlands (Shannon index = $2 \pm 0.2$) followed by wooded grasslands ($1.5 \pm 0.2$).

### 3.3 Drivers of non-cultivated tree carbon stock per land use

Drivers in this article are factors that contribute to carbon stock variability. Tree carbon stock (VC) varied according to land use (table 1). Bushlands had the highest carbon stocks ($6.0 \pm 2.4$) followed by the forests ($4.8 \pm 0.4$) and the wooded grasslands ($4.3 \pm 2.6$). Non-cultivated trees from perennial croplands and annual croplands had the lowest stocks of carbon. The total carbon stock stored by the non-cultivated trees was around $26.8 \pm 5$ k t for the 100 km² area.

All the tree parameters measured (tree density, species diversity, trees height and tree diameter at breast height) had effect on tree carbon stock with different degrees (figure 2). Tree density had the highest effect with a path-standardized coefficient of 0.66. Tree diversity had a direct effect and an indirect effect with coefficients of 0.27 and 0.52 respectively. Tree height had a small and direct effect (0.26) and tree diameter, which strongly affected tree height, had the smallest effect (0.22).

### 3.4 Soil organic C and N concentrations per soil depth and per land use

The concentration of SOC decreased from 0 to 110 cm soil depth in all the different land uses (figure 3a). Soil organic carbon concentration in the 0- 20 cm depth was similar in all land uses. SOC concentration was significantly different between land



uses only from 20 to 110 cm. Soil under perennial crop had higher SOC concentration ($7 \pm 1$ g C kg$^{-1}$) compared to the wooded grasslands ($4 \pm 0.4$ g C kg$^{-1}$) in the $20 - 50$ cm horizon. SOC were significantly higher in perennial ($5 \pm 1$ g C kg$^{-1}$) and annual croplands ($5 \pm 1$ g kg$^{-1}$) than in wooded grasslands ($2 \pm 0.4$ g C kg$^{-1}$) at $50 - 80$ cm. Perennial croplands ($3 \pm 0.2$ g C kg$^{-1}$) and

bushlands ($3 \pm 0.3$ g kg$^{-1}$) had the highest SOC while wooded grasslands had the lowest SOC concentration at $80 - 110$ cm. Soil total nitrogen (TN) also decreased with soil depth (figure 3b). As with soil organic carbon, wooded grasslands had the lowest TN concentration from 0 to 110 cm of soil depth. The highest TN concentration was found in bushlands and forest ($1.1 \pm 0.1$ g N kg$^{-1}$) at $0 - 20$ cm. However, at $20 - 50$ cm and $80 - 110$ cm the highest TN values were found in annual croplands, perennial croplands and bushlands. From $50 - 80$ cm, perennial and annual croplands had the highest TN concentration ($0.4 \pm$

$0.04$ g kg$^{-1}$ and $0.4 \pm 0.04$ g kg$^{-1}$).

**3.5 Soil organic C and N stocks per soil depth and per land use and their drivers**

Soil organic carbon (SOC) stock decreased from $0 - 20$ cm to $80 - 110$ cm for all land uses similarly to SOC concentrations (figure 4a). SOC stock was higher in forests, bushlands, and perennial croplands ($42 \pm 9$ t C ha$^{-1}$, $38 \pm 12$ t C ha$^{-1}$, $25 \pm 2$ t C

ha$^{-1}$) than in wooded grasslands ($17 \pm 2$ t ha$^{-1}$) at $0 - 20$ cm. The same trend was observed at $20 - 50$ cm. From 50 to 80 cm, SOC stocks were similar in the land uses while at $80 - 110$ cm wooded grasslands ($9 \pm 1$ t ha$^{-1}$) had the lowest SOC stock and bushlands ($14 \pm 1$ t ha$^{-1}$), perennial croplands ($14 \pm 1$ t ha$^{-1}$) had the highest SOC stock. The total SOC stock from 0 to 110 cm in the 100 km² landscape was $349.6 \pm 6$ k t which was 13 times higher than the VC stock.

TN stocks are shown in figure 4b. Bushlands ($2.6 \pm 0.3$ t N ha$^{-1}$) and perennial croplands ($1.6 \pm 0.1$ t ha$^{-1}$) had higher TN stock than wooded grasslands ($1.4 \pm 0.1$ t ha$^{-1}$) at $0 - 20$ cm. Similarly, to SOC stock, at $80 - 110$ cm wooded grasslands ($0.7 \pm 0.1$

t ha$^{-1}$) had the lowest TN stock and bushlands, perennial and annual croplands had the highest TN stocks. The total TN stock from 0 to 110 cm in the 100 km² was about $27.1 \pm 3.8$ k t.

Different drivers affected SOC stock at the different depths. The effect of soil clay content on SOC stock was 0.55 at $0 - 20$ cm (figure 5) and clay content effect increased with soil depth (figures S6, S7 and S8 in supplementary material). Clay content

was the principal driver controlling SOC stock at $50 - 80$ cm. Soil coarse content affected SOC stock only at $20 - 50$ cm and $80 - 110$ cm. Slope, erosion and fire had negative and indirect effect on SOC stock by decreasing soil clay content.

The path analysis showed that TN stock was affected by the same factors as soil organic carbon stock (figure S9 in supplementary material).

**3.6 Relationships between SOC, N, and VC stocks**

VC and SOC stocks were not correlated with each other when considering all data. However, significant relationships were found between SOC stock and VC stock in annual croplands, perennial croplands and grasslands (figure 6). No significant relationship was observed in wooded grasslands. A significant relationship between SOC and VC stocks was also observed in





the bushlands with no depth restrictions before 110 cm. However, in the presence of soil depth restriction no significant relationship was found between SOC and VC stocks for bushlands.

## 255 **4 Discussion**

We discuss first the variation of uncultivated tree C stocks and soil C and N stocks, then the variables driving these stocks and finally the relationships between C stocks in uncultivated trees and in soils to answer the hypothesis made at the beginning of the paper.

### 4.1 Variation of the stocks of C in uncultivated trees, in soil C and in soil N in the different land uses

The higher VC stock in the forest and wooded grasslands compared to the annual croplands is attributed to the high tree density and diversity in the former land uses. Similar results have been reported in the Sudanese ecological zone of Burkina Faso (Dayamba et al., 2016; Dimobe et al., 2019).

Soil organic carbon stocks also varied according to the land use. Perennial croplands had the highest SOC stock at $0 - 50$ cm and $80 - 110$ cm, probably due to the replenishment of soil SOC by leaf litter and roots of cashew trees. This result is in
agreement with Poeplau and Don (2015) who found high SOC stocks in agro-ecosystems with perennial cover crops in different soil and climate conditions. Bello et al. (2017) found lower SOC stock at $0 - 20$ cm in cashew plantation in the transition zone in Benin compared to our results probably due to the difference in cashew density and plantation age and soil texture.

The low SOC stock in the wooded grassland may be explained by the low clay content in these land uses located in steepest
slope and subjected to erosion and fire (table 2).

The high TN stock observed in bushlands and perennial croplands at $0 - 50$ cm and forest, perennial and annual croplands, bushlands at $80 - 110$ cm may be due to the high clay content, but also to the high tree diversity in bushlands. Indeed, according to Haghverdi and Kooch (2019) higher trees diversity enhance the level of organic carbon and particulate organic nitrogen in soils. The effect of tree diversity on TN stocks were also highlighted in various biophysical conditions. For instance, Li et al.
(2019) in a desertification experiment site in China observed low TN stock (0.31 - 0.89 t ha⁻¹) at $0 - 20$ cm depth in mono-cropping of *Caragana intermedia* plantations. The soil clay and carbon contents and bulk density are other factors that may explain differences in TN stocks. Indeed, Saiz et al. (2012) and Zhong et al. (2018) showed in West Africa and China respectively that higher clay content and higher bulk density lead to higher SOC stock thus higher TN stock.

Our results suggest that perennial crops cultivation and bushlands preservation in the forest – savanna transition zone of
northeast Tiéningboué should be recommended for soil organic carbon preservation



## 4.2 Drivers of the stocks of C in non-cultivated trees and in soils, and of the N soil stock at landscape level in a forest – savanna transition zone

Our results showed that VC stock is mainly driven by tree density and diversity, suggesting that increasing the number of trees stands per unit area and also the number tree species would contribute to increase VC stock. Some authors have also shown a positive correlation between the Shannon diversity index and above- and belowground biomass carbon in protected savanna of Burkina Faso (Dayamba et al., 2016; Dimobe et al., 2019) and in the Guinean savanna ecosystem in northern Sierra Leone (Amara et al., 2019). On the other hand, a negative correlation between biomass carbon and plant diversity has been reported in a subalpine coniferous forest and in community forests (Zhang et al., 2011; Aryal et al., 2018), while no correlation was found in a farmland in Kenya (Henry et al., 2009). These contrasting results can probably be explained by the difference in tree diversity between the different areas (Markum et al. 2013, Dimobe et al. 2018).

We found that soil clay content is the principal factor controlling SOC stock, which is in agreement with earlier observations by Feller and Beare (1997) and Hounkpatin et al. (2018). Our result not only showed a negative effect of the slope on SOC stock storage but also demonstrated that slope and erosion affect SOC stock indirectly through decreasing soil clay content. Higher slope and higher erosion might also contribute to higher transportation of fine particles rich in C, N and clay to the bottom of the slope. Thus, controlling the causes of erosion could help reducing clay losses and maintaining SOC stocks. However, this erosion can also contribute to an enrichment in clay and organic matter of lands located at the bottom of the slope and thus serving as a carbon sink where decomposition rate will be reduced (Van Oost et al. 2007). Other factors reported as important contributors to SOC storage such as parent material, soil type and climate conditions (Saiz et al., 2012, Barré et al., 2017, Zhu et al., 2019) have not been considered in our study and need further investigations to fully understand SOC variations on a larger scale.

## 4.3 Correlation between non-cultivated trees carbon stock and soil organic carbon stock in different land uses

Our starting hypothesis cannot be rejected as indeed we observed different types of relationships between the C stocks in uncultivated trees and soil organic C stocks for different land uses. These relations were positive for the land uses annual crop land, perennial crop land, grassland, and bushland without depth restrictions in the 0-110 cm depth. On the contrary no relation was found between these two types of stocks in the land uses wooded grassland and bushland with a soil depth restriction in the 0-110 cm depth. These results are discussed thereafter in relation to the empirical land use dynamic model that we assumed happened at this site during the last 50 years which is presented in figure 7.

The land use dynamic model was constructed based on our LDSF data, literature data and own observations. The land use ages are estimations based on discussion with stakeholders. Old bushlands are cleared for annual crops cultivation during at most four years. Annual crops are then replaced by perennial crops which can occupy the soil for more than 35 years. Only 1 % of grasslands located in the lowlands are now cropped with rice. Wooded grasslands and most of grasslands are not converted to crop lands. These land uses are dedicated to other purposes such as grazing, hunting, collecting medicinal plants and timber and firewood collection. Whereas forest was covering the vast majority of the region 100 years ago (Benveniste, 1974, Bockel



et al., 2021), there is only 1.4% of the area now covered with forest. The remaining forests are close to the villages and used for hunting, collecting medicinal plants and for cultural and religious practices. This pattern suggests that land uses have been chosen by farmers according to land morphology and soil characteristics. Indeed, wooded grasslands and grasslands being located on the steepest slopes are low in organic matter and nutrients and thus not suitable for cropping. Bushlands on the contrary are preferably used by farmers for annual cropping given their high clay, SOC and TN contents and their rather flat relief. Kassi et al. (2017) also observed that farmers preferably use those bushlands for cropping in the centre of Côte d'Ivoire. The forest growing in past on what is now a bushland with soils without depth restriction has probably been very productive because of the favourable soil conditions. We assumed that it largely contributed to the current SOC stock which itself continues to support the growth of current uncultivated trees, explaining the positive relation between SOC and VC stocks not only in the bushland, but also in the annual crop and perennial lands derived from bushlands. Similarly, these are soil properties that explain the lack of relation between VC stocks and SOC, in bushland with depth restriction, limiting tree growth and in the wooded grassland which are located on steep land, containing low clay content and are impacted by fire, grazing and erosion.

**5 Conclusion**

Soil and tree characteristics were measured and drivers affecting VC stock, SOC and TN stocks variability were investigated in various land uses of a forest – savanna transition zone. We found that soils in wooded grasslands and grasslands had low clay content, were on steeper slopes, and impacted by erosion, grazing and fire. These characteristics contributed to the fact that SOC and TN stocks in wooded grasslands were lower compared to perennial croplands (cashew plantations) and bushlands especially at $0 - 20$ cm and $80 - 110$ cm soil depth probably due to leaves and roots litter. VC stock was mainly affected by tree density and tree diversity, suggesting that increasing the number of trees stands per unit area and also the tree diversity would contribute to increase VC stock. SOC stock and TN stock were mainly affected by clay content followed by bulk density. Our results not only showed a negative effect of the slope on SOC stock storage but also demonstrated that slope and erosion affected SOC stock indirectly through soil clay content. Thus, controlling the causes of erosion could help reduce clay losses and therefore maintain or increase SOC stock at the site of measurement. This study suggests that VC stock positively increase SOC stock, and this effect is more pronounced in deep soils with high clay content. Our results indicate that soil properties and morphology determined land use and perennial crops cultivation and bushlands preservation in the forest – savanna transition zone of northeast Côte d´Ivoire can contribute to increase SOC and TN stocks.

**Funding**

This work was funded by the Swiss Program for Research on Global Issues for Development (SNF/SDC) through the YAMSYS project (SNF project number: 400540_152017 / 1). ILBOUDO also beneficiated a scholarship from Swiss Federal Commission for Scholarships for Foreign Students.



**Acknowledgments**

We thank the YAMSYS team and collaborators, the geo-science laboratory team at ICRAF Nairobi, the LDSF team at ICRAF
Côte d'Ivoire, the plant nutrition group at ETHZ and the agro-ecosystem group at ETHZ. We are also grateful to all farmers
and authorities at Tiéningboué, Côte d'Ivoire.

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



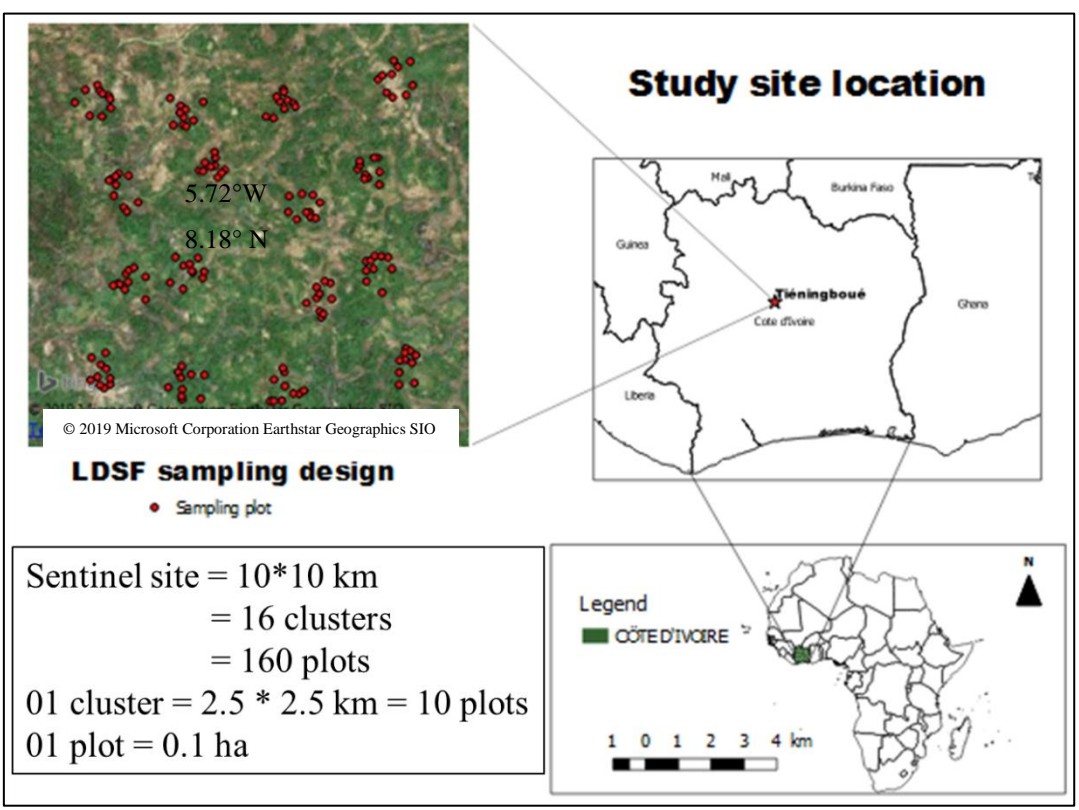


Figure 1: Study area location in Tiéningboué, West Africa and Land Degradation Surveillance Framework sampling design.






Table 2: Vegetation characteristics of the different land uses in a forest – savanna transition zone, Tiéningboué, West Africa.

| | Forest | Bushland | Wooded grassland | Grassland | Perennial cropland | Annual cropland |
|---|---|---|---|---|---|---|
| Surface area (%) of the 100km² | 1.4 *(n=2)* | 12.5 *(n=18)* | 31.2 *(n=45)* | 6.9 *(n=11)* | 24.3 *(n=35)* | 23.6 *(n=34)* |
| DBH (cm) | $61.8 \pm 2.0^{abc}$ | $56.1 \pm 16.1^{bc}$ | $48.1 \pm 17.6^{c}$ | $56.1 \pm 29.1^{bc}$ | $113.0 \pm 64.0^{a}$ | $63.5 \pm 34.8^{b}$ |
| TH (m) | $8.7 \pm 0.1^{abc}$ | $8.5 \pm 1.7^{abc}$ | $6.7 \pm 1.8^{c}$ | $6.9 \pm 2.5^{bc}$ | $10.6 \pm 3.4^{a}$ | $8.9 \pm 2.6^{ab}$ |
| TD (nb ha⁻¹) | $283 \pm 4^{abc}$ | $452 \pm 210^{a}$ | $402 \pm 190^{a}$ | $154 \pm 92^{b}$ | $15 \pm 15^{c}$ | $40 \pm 34^{bc}$ |
| Di (index) | $1.9 \pm 0.1^{abc}$ | $2.0 \pm 0.2^{a}$ | $1.7 \pm 0.3^{b}$ | $1.5 \pm 0.2^{bcd}$ | $1.2 \pm 0.3^{d}$ | $1.5 \pm 0.4^{c}$ |
| VC (t ha⁻¹) | $4.8 \pm 0.4^{abc}$ | $6.0 \pm 2.4^{a}$ | $4.3 \pm 2.6^{b}$ | $1.8 \pm 1.5^{bc}$ | $0.5 \pm 0.4^{c}$ | $0.9 \pm 1.0^{c}$ |
| DS | *Sterculia tragacantha* | *Margaritaria discoidea* | *Uapaca togoensis* | *Terminalia scimperiana* | *Parkia biglobosa* | *Margaritaria discoidea* |

Mean ± SD, DBH= mean tree diameter at breast height in cm, TH= mean tree height in m, TD= tree density (tree number per ha), Di= Shannon index of tree species diversity, VC= tree carbon stock (above and belowground) ton per ha, DS= dominant species; means were compared using Tukey test in lsmeans package R software, means with same the letter are not significantly different between land uses, n= the number of sampled plots.



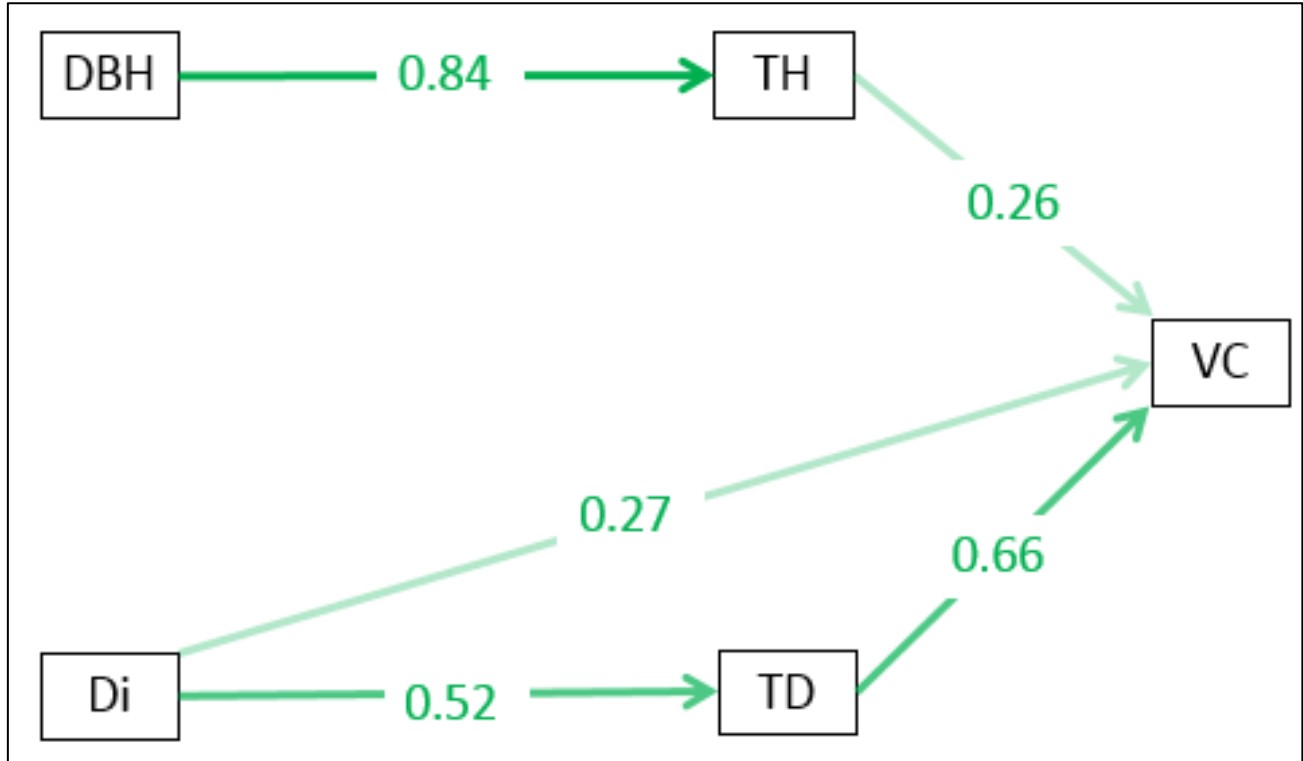

VC= Carbon stock by non-cultivated trees, TD= Trees density, Di= Trees species diversity, TH= Tree height, DBH= Tree diameter at breast height, single-headed arrows indicate direct causal relationships. Numbers are path coefficients and numbers within single-headed arrows indicate proportion of total variance explained for each variable. Green arrows represent positive effect and red arrows represent negative effect, the darker is the color, the stronger is the effect.

Figure 2: Drivers of non-cultivated tree carbon stock at landscape level in a forest – savanna transition zone in West Africa.






Table 2: Soil clay content (g kg$^{-1}$) per soil depth (cm) and land use in forest – savanna area in West Africa.

| Depth | Forest | Bushland | Wooded Grassland | Grassland | Perennial cropland | Annual cropland |
|---|---|---|---|---|---|---|
| 0 – 20 | 354.4 ± 19$^{abc}$ | 316.6 ± 51$^{ab}$ | 240.9 ± 82$^{a}$ | 243.9 ± 73$^{bc}$ | 352.3 ± 2$^{a}$ | 326.5 ± 2$^{ab}$ |
| 20 – 50 | 432.6 ± 144$^{abc}$ | 369.9 ± 102$^{ab}$ | 258.7 ± 130$^{c}$ | 252.0 ± 124$^{bc}$ | 395.4 ± 4$^{a}$ | 368.6 ± 5$^{ab}$ |
| 50 – 80 | 541.5 ± 218$^{abc}$ | 489.5 ± 190$^{ab}$ | 289.9 ± 180$^{c}$ | 275.0 ± 157$^{bc}$ | 483.6 ± 3$^{a}$ | 552.3 ± 4$^{a}$ |
| 80 – 110 | 559.5$^{abc*}$ | 492.3 ± 180$^{ab}$ | 284.2 ± 199$^{c}$ | 327.7 ± 250$^{bc}$ | 568.0 ± 4$^{a}$ | 468.7 ± 4$^{ab}$ |
| 0 – 110 | 459.5 ± 139$^{a}$ | 400.2 ± 146$^{a}$ | 276.2 ± 153$^{b}$ | 273.8 ± 148$^{b}$ | 457.2 ± 152$^{a}$ | 404.5 ± 153$^{a}$ |

Mean ± standard deviation, means with the same letters are not significantly different between lands uses, means were compared with the test of Tukey, * one
soil sample









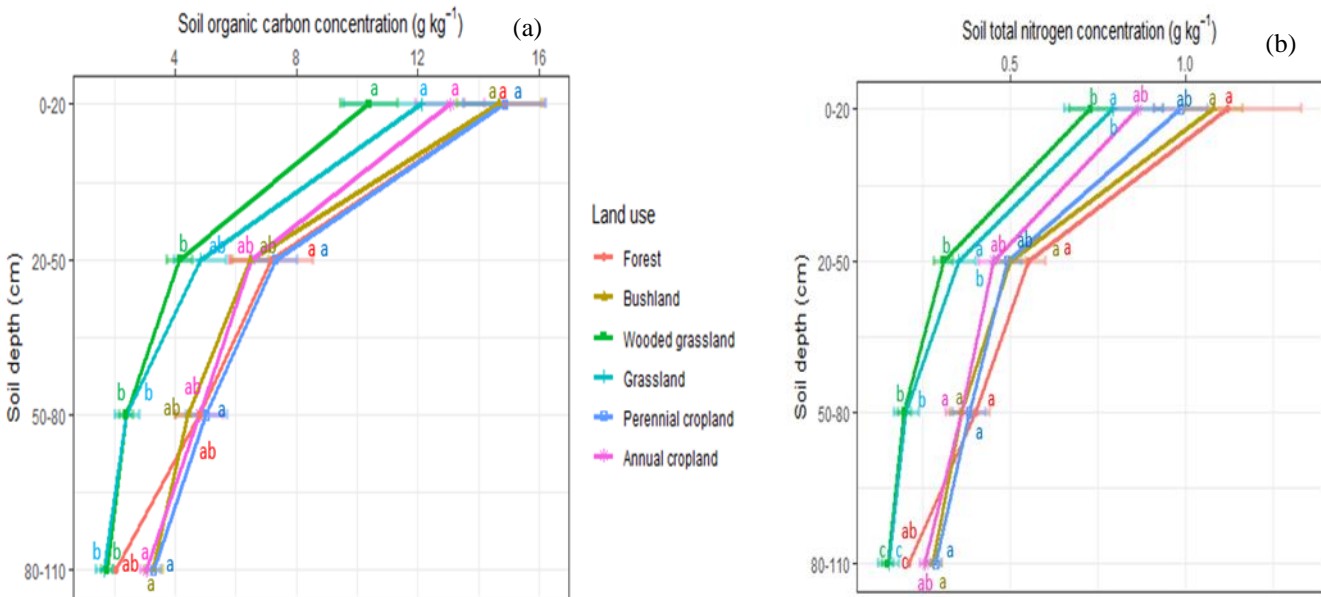

Mean with standard error, means with the same letters are not significantly different between land uses, means were compared using the test of Tukey in lsmeans package.

Figure 3: (a) Soil organic carbon concentration and (b) soil total nitrogen concentration variations with soil depth at landscape level in West African forest savanna transition.







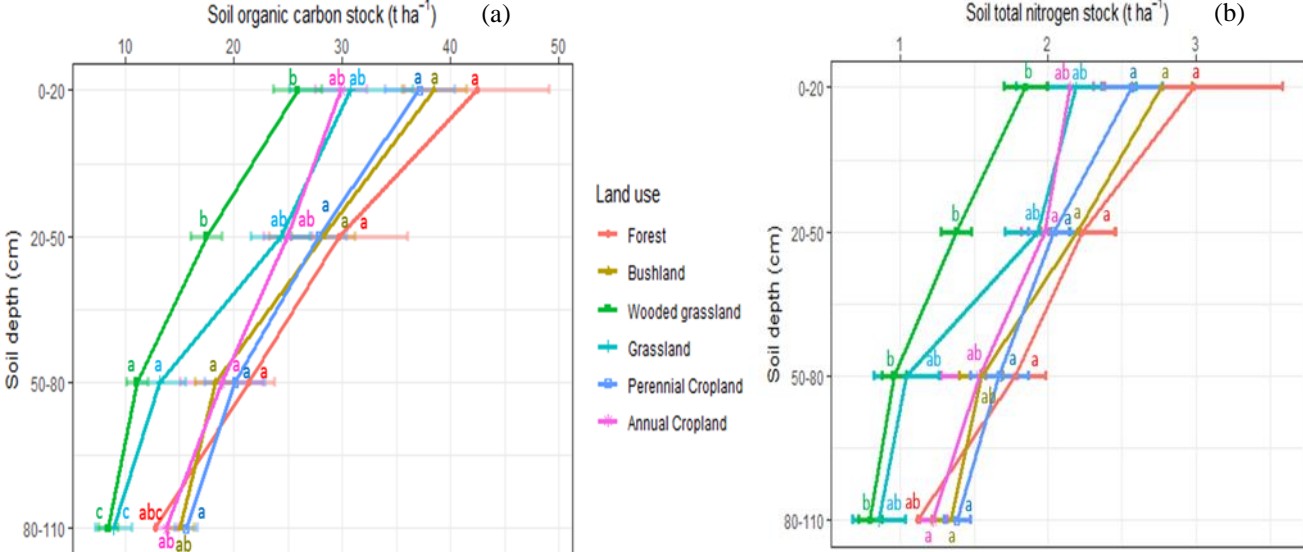

Mean with standard error, means with the same letters are not significantly different between land uses, means were compared using the test of Tukey in lsmeans package.

Figure 4: (a) Soil organic carbon stock and (b) soil total nitrogen stock variations with soil depth in different land uses in West African forest savanna transition.







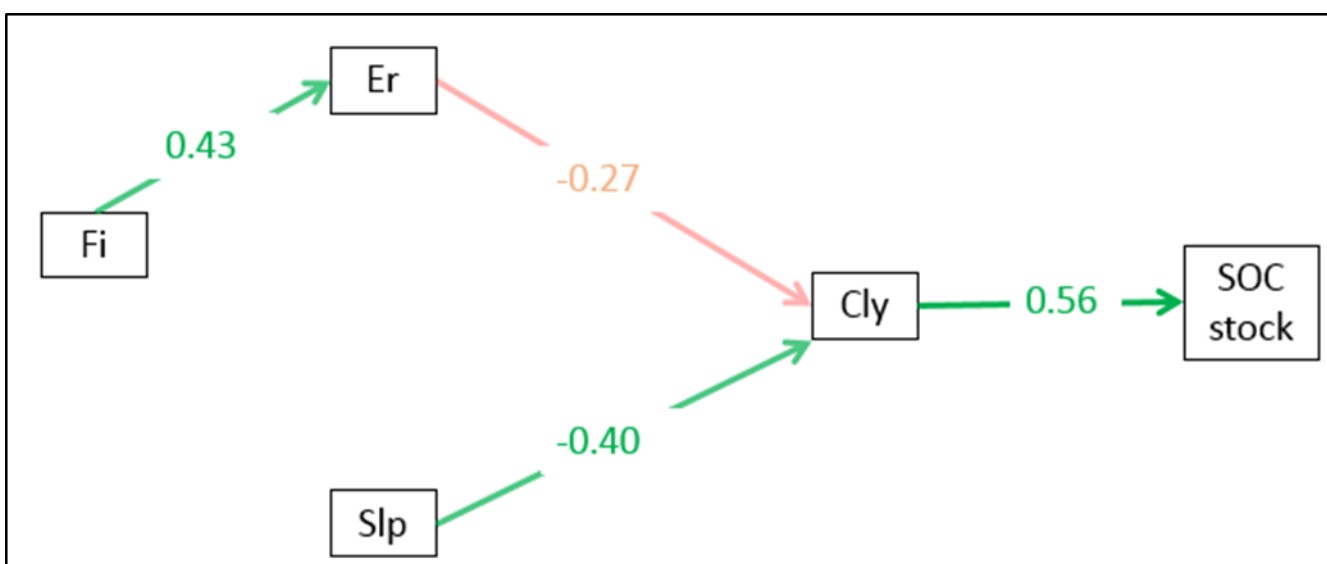

SOC stock= soil organic carbon stock, Cly= soil clay content, Er= erosion impact, Fi= fire impact, Slp= slope, single-headed arrows indicate direct causal

relationships. Numbers are path coefficients and numbers within single-headed arrows indicate proportion of total variance explained for each variable. Green

color represents positive effect and red color represents negative effect.

Figure 5: Soil organic carbon stock factors determined at 0-20 cm by path analysis at landscape level in West African forest

savanna transition.









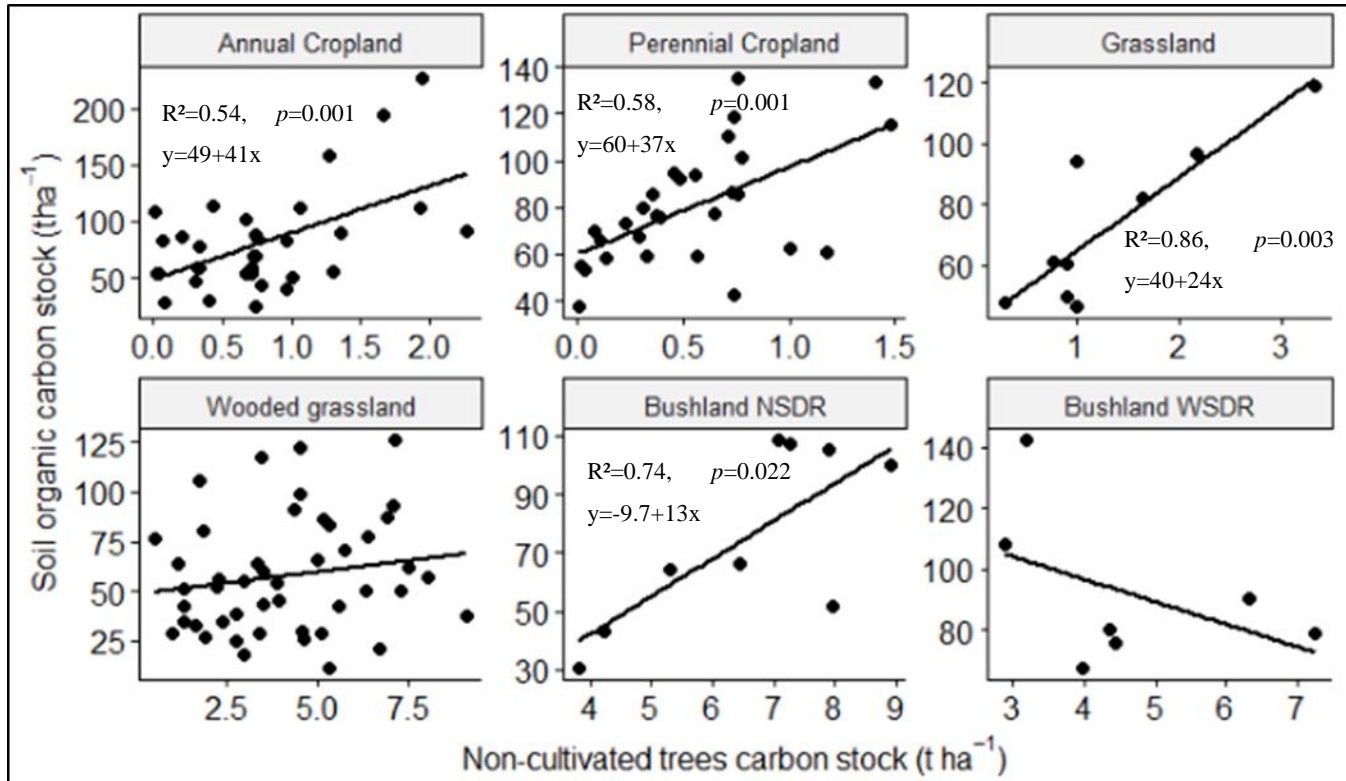

NSDR = Bushlands with no soil depth restriction till 110cm, WSDR = Bushlands with soil depth restriction till 110cm, forests are absent in this figure because
they were only two forests, the interval of confidence is between 2.5 to 97.5 for the linear regressions.

Figure 6: Relationship between soil organic carbon stock and non-cultivated tree carbon stock per land uses at soil depth 0 -
110 cm in a forest savanna transition zone in West Africa








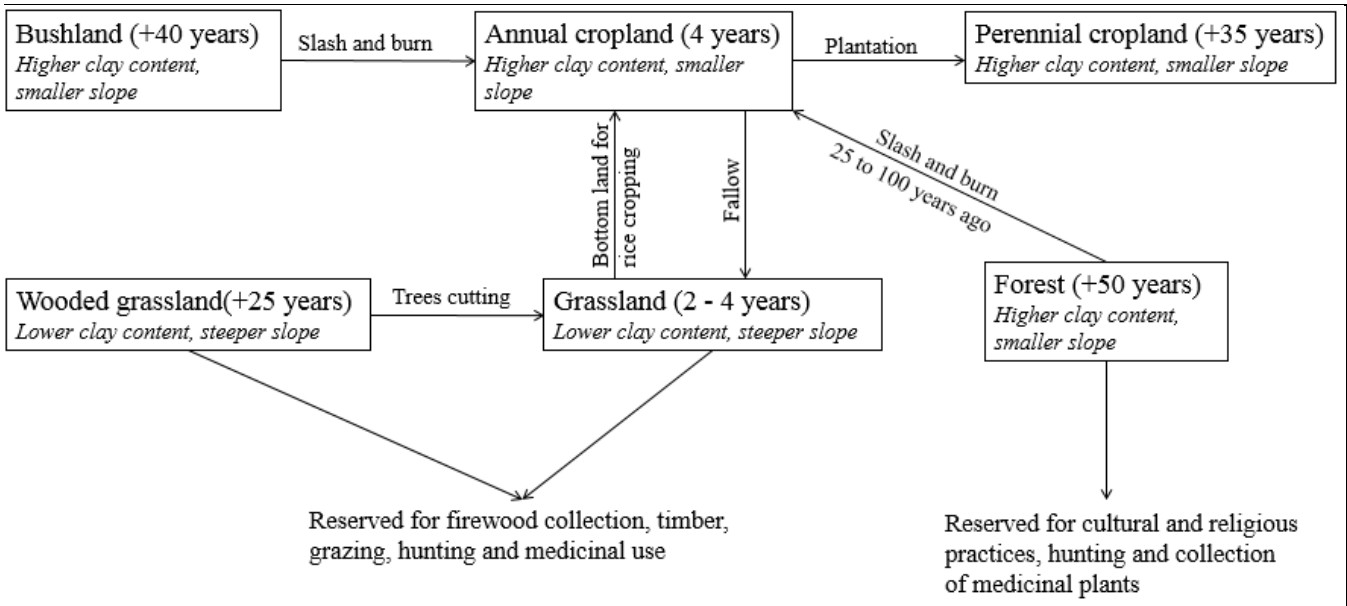

Figure 7: Land use dynamic model in 2015 related to soil characteristics in a forest savanna transition zone in West Africa.
Italic text represents the characteristics of the land use compared to the other land use.