# Peer review of "Relationship between the stocks of carbon in non-cultivated trees and soils in a West-African forest-savanna transition zone"

_EGUsphere, 2022_

## Author Comment (AC1)

**Authors responses to comments of Referee 1**

Dear Referee,

We thank you for your precious time in reviewing our paper **" Relationship between the stocks of carbon in non-cultivated trees and soils in a West-African forest-savanna transition zone (MS No.: egusphere-2022-209)".** We appreciate your valuable comments that will help us improve our manuscript.

The authors have carefully considered the comments and tried our best to provide the below point-by-point responses for your comments and questions. Thank you very much for your interest in this manuscript.

Sincerely,

The authors

| I. Referee #1 | |
|---|---|
| **General comments** | |
| **Comments / Question** | **Our Reply** |
| **R1Q1.** In spite of the fact that this aim is somewhat apparent in the paper's title (except the land use), it is extremely troubling that the paper fails to frame and thus address any specific problem regarding this relationship and to set forth a hypothesis for testing | Climate warming is actually a global problem all over the world. Also, carbon is a primordial element in soil fertility. One of the solutions to this global problem is carbon storage. Our study contributed in consolidating knowledge of processes of carbon storage. |
| **R1Q2.** The impact of land use (different configurations) on soil organic C stocks and their relationship has been adequately studied, so it is important for readers to know the gaps and how they can be resolved. | There are not many studies that are done on this subject especially in west Africa. Besides as referee 2 mention it R2Q4. |
| **R1Q3.** This may be due, in part, to the weak definition of non-cultivated | In our context, farmers open new farmlands by slashing and burning forests, bushlands or wooded grasslands (see photo below). In this process they spare few trees that are important to them socially, |

| trees and its conceptual relation to land use. I find it challenging that such a class of trees are located even in forests and bushlands. | economically and / or nutritionally as we mention in reply of R2Q8. These spared trees are the "non-cultivated trees" in the annual and perennial croplands in our study.

In order to have the same given name, we used the term "*non-cultivated trees*" for all land uses.

So, in forests and bushlands, all trees that were measured were considered as non-cultivated trees. |

03/06/2015 09:40

| R1Q4. Were the forests plantations? | For our study context, forests are natural vegetation that are not disturb for more than 50 years by human actions like cropping, building. There are naturally generated without any tree plantation. There are especially used for traditional rites. The below photos show how the forest looked like. |

23/06/2015 14:10

| | |
|---|---|
| **R1Q5.** What is the area of influence of these non-cultivated trees in cultivated and non-cultivated fields? | In this study, the area of influence of non-cultivated trees is the LDSF plot (0.1ha). |
| **R1Q6.** Were the soils sampled within this area of influence accordingly? | Soil samples were taken in the center of each plot. |

| | |
|---|---|
| **R1Q7.** There seems to be an unjustified attempt throughout the paper (e.g. the captions of the table and figures) to generalize its empirical findings to the entire West African region | This is only describing the case of a forest savanna transition zone in west Africa. We will check the paper not to generalize its findings to the entire West Africa. |
| **R1Q8.** It is difficult to argue that this paper provides any new knowledge or makes any contribution to the current body of knowledge, especially on this fascinating topic of terrestrial C dynamics. | This study is a data base to study carbon dynamics. To achieve this objective, the same survey will be done periodically (every 10 years) in the same LDSF site.

The term "forest savanna transition area" may cause some confusion in understanding it. In our study site, the savanna zone did not succeed gradually to the forest zone. It is like a buffer zone see photo below) where forest and savanna zones were all together without a straight line between them. |

**Specific comment in Abstract**

**R1Q9.** There is no explicit statement on the objectives/aims/questions of the paper, and hence how they were

In this study, our general objective was to contribute to reduce climate warming by carbon storage.

Our specific objectives were to:

- Determine principal drivers that influence carbon storage
- Determine the relationship between the stocks of carbon in non-cultivated trees and soil

| | |
|---|---|
| achieved/answered. Also, be consistent on the study area as this is an empirical work that does not cover the whole forest-transition zone of West Africa. | |
| **Specific comments in introduction** | |
| **R1Q10.** There is complete lack of context and too many generalizations without a critical assessment of the current state of literature. I suggest to rewrite the introduction providing clear context on the research problem by engaging the contemporary literature on terrestrial carbon dynamics. This can help readers to appreciate the exact contribution of this work. | Ok, we will improve the introduction but as mention earlier, this is not a carbon dynamics study. |
| **R1Q11.** Against the claims of the paper, there is no testable hypothesis provided. In line 74-75, the | It may be a fact for some regions but this fact needs to be confirmed in others regions like African regions. |

| | |
|---|---|
| hypothesis of the paper is given as "…we hypothesized that the relationship between the stock of soil organic C and the stock of C in the uncultivated will not be identical in all land uses", which is a statement of fact and not a hypothesis. It is indeed a historical fact that land use influence above and below ground C stocks. Also, this hypothesis is unfalsifiable | |
| **Specific comments in methods** | |
| **R1Q12.** I suppose the lack of context and a testable hypothesis in the introduction also affected this methods section. This is because, while it is clear that the sampling design of this work follow the LSDF design, it is largely unexplained as to how the aims of this work align with those of the LSDF, given that the | This work still provides a baseline information. The little difference in our study is that we tried to see the relationship between trees carbon stock and soil carbon stock with the data. We used LDSF design in to monitor carbon dynamics with time. With LDSF design, the coordinate of every sampled plot are registered and at any time we can come back for the same sampling. |

| | |
|---|---|
| original design was to provide baseline information for land degradation processes. | |
| **R1Q13.** Also, what are some of the unique features of the selected area that makes it useful and representative for the aims/objectives of the study. Which of these features are generalizable and which are not? | The selected area is a forest-savannah transition zone. Thus, we observed different type of vegetation, land uses and soil properties. Yam were also the main crop among the annual crops. Those features are generalizable except the cultivated crops that vary with farmer preferences. |
| **R1Q14.** In line 95, "The LDSF as it uses a nested hierarchical sampling design allows for the development of predictive models that has a global coverage without changing the local relevance", please explain | Data collected with LDSF design are hierarchically nested (figure 1). Data from many LDSF sites can be used to develop predictive models with global coverage while maintaining local relevance. Local differences in site level may not affect the predictive model.  |
| **R1Q15.** It is inadequate to state that "land use classification was done using a simplified version of the FAO | According to Vågen, Winowiecki and Tondoh (2013), land cover is recorded in all plots using a simplified version of the FAO Land Cover Classification System (LCCS), which was developed in the context of the FAO-AFRICOVER project (http://www.africover.org). In addition, vegetation is |

| | |
|---|---|
| Land cover classification system", please explain how and why it was implemented in this study? As it is, I fail to understand why a global classification system is used for such a small local study. | classified according to White, 1983. Also, scores are made of "impact on habitat", adapted from Royal Botanic Gardens, Kew ([http://www.kew.org](http://www.kew.org)). |
| **R1Q16.** Line 98-99, in which way is the contribution to SOC of annual crops different from that of perennial crops? | Perennial crops may contribute more to SOC stock in term of C quantity compared to annual crops due to the fact that they stay longer in the plot. |
| **R1Q17.** Line 99-100, please explain how the data on the impacts of erosion, fire and grazing were collected? Also, be specific on the topographic features that were collected and how they were collected | Data on the impacts of erosion, fire, grazing and topographic position (figure 4) were collected by visually inspecting the area surrounding the plot. Erosion, fire and grazing impacts were recorded (figure 2) according the severity (from 0 = none to 3= severe).  Figure 2: impact on habitat recording sheet (Vågen, Winowiecki and Tondoh, 2013) In each sub-plot (1, 2, 3 and 4), signs of visible erosion were also recorded and classified as rill, gully and sheet (figure 3).  Figure 3: Signs of visible erosion recording sheet (Vågen, Winowiecki and Tondoh, 2013) |

Figure 4: Topographic positions (Vågen, Winowiecki and Tondoh, 2013)

| | |
|---|---|
| **R1Q18.** What informed the reason for counting only trees that had a diameter of >2.5 cm and a height of >1.5?

And why was the radius for tree data collection different in the annual croplands compared to the others? | We defined these dimensions using many works as Rondeux (1978), Rondeux (1999).

The radius for tree data collection was different in the annual croplands compared to the others because there few (e.g. 18 trees per plot of 0.1ha) trees in the annual croplands (see photo below) compared to others like bushlands (e.g. 107 trees per plot of 0.1ha).

However, we made sure that this difference does not affect our statistical analysis. Trees data from croplands did not vary according to the radius. We will present the same radius for all land uses in order to avoid confusion. |

| | |
|---|---|
| **R1Q19.** In section 2.3, line 122-124, was the pH of the soils measured? Because it is inadequate to assume that the values for some soils in other | The pH of these LDSF sampled soils were not measured. However, for other study of my PhD, 38 soil samples were taken in yam fields of the LDSF site at 0-30 cm soil depth. The pH of yam fields soil samples was measured. The results showed that 87% of the soil samples presented a mean pH was 6.05. The pH varied from 5.2 to 6.8. |

| | |
|---|---|
| parts of West Africa will automatically apply to your own soils. | Besides others study like N'Dri and André (2011) found acidic pH from soils of central Ivory Coast. |
| **R1Q20.** • Line 125-126, which specific packages in R (it is important for developers of such packages to be acknowledged whenever possible) | We used many R functions such as *plot ()* function to visualize and identify bad spectra. *Which ()* function was used to exclude bad spectra and outliers, *t ()* function for first derivative. With the *calibrate ()* function, we developed *PLS models* of total C, total N, clay, silt and sand content using both spectral and chemical analyses data of the reference samples. *Round ()* function was used to select the reference samples. Maybe we should mention "R functions" instead of "R packages" in the manuscript. |
| **R1Q21.** Please add the information on the calibration plots (from the spectroscopy data) to the main paper, and not in the supplementary. | Ok. |
| **R1Q22.** How many soil samples were collected in total, and how many constituted the 15%? How representative was this 15% regarding the feature space? | In total, 594 soil samples were collected, thus 90 soil samples constituted the 15%. The 15% were selected randomly with kenard stone among the total soil samples. This percentage was determined to fit the developed models for our LDSF site. |
| **R1Q23.** Line 125-132, this whole paragraph needs to be re-written to improve clarity. | Ok, we will clarify this paragraph. |
| **R1Q24.** Line 140-144, please clarify and explain the basis for using bulk | We used soil bulk density (1.4) from Hounkpatin et al. (2018) which was done in Dano in Burkina Faso. This bulk density result is similar to bulk density of Tieningboué shown in soilgrids data. Bulk density |

| | |
|---|---|
| density values from Burkina Faso when the soils and the ecological conditions are so different. One would assume values from other neighboring countries with similar agroecological conditions might be applicable. | measured by Hounkpatin et al. (2018) was also similar to bulk density measured by Kassi et al (2017) in the center of Côte d'Ivoire. |
| **R1Q25.** It is largely unclear to me why the path analysis was used especially as there is no hypothesis to be tested. Obviously, the path analysis is a statistical analysis and so it should be part of the same section | For the path analysis, we prepared a conceptual model with many assumptions presented in the figure S5.

Ok, we will put this paragraph in the statistical analysis paragraph. |
| **Specific comments in results and discussions** | |
| **R1Q26.** Sections 3.1 and 3.2 are quite basic information that should have been provided in the methods section or seem to be irrelevant to the work. Figure S1 is a table, not a figure. There is no table 1 in line 175. | These informations are results of our survey and are important to understand and explain the other results.

Ok, we will change "Figure S1" to "Table S1" |

| | |
|---|---|
| **R1Q27.** As to be expected the main finding was that SOC stocks vary with land use… of course they do. In Line 263-268, the argument to support the reason why perennial crops had on average higher SOC is weak and contradictory, please check. | *"Perennial croplands had the highest SOC stock at 0 – 50 cm and 80 – 110 cm, probably due to the replenishment of soil SOC by leaf litter and roots of cashew trees. This result is in agreement with Poeplau and Don (2015) who found high SOC stocks in agro-ecosystems with perennial cover crops in different soil and climate conditions. Bello et al. (2017) found lower SOC stock at 0 – 20 cm in cashew plantation in the transition zone in Benin compared to our results probably due to the difference in cashew density and plantation age and soil texture."* |
| **R1Q28.** • Bizzare conclusion: " Our results suggest that perennial crops cultivation and bushlands preservation in the forest – savanna transition zone of northeast Tiéningboué should be recommended for soil organic carbon preservation". It is quite difficult to draw such a conclusion from a single empirical study, please revise | We made this suggestion base on our results but, indeed, our results still needs to be confirm by further studies. |

---

## Author Comment (AC2)

**Author responses to comments of Referee 2**

Dear Referee,

We thank you for your precious time in reviewing our paper " **Relationship between the stocks of carbon in non-cultivated trees and soils in a West-African forest-savanna transition zone (MS No.: egusphere-2022-209)".** We appreciate your valuable comments that will help us improve our manuscript.

The authors have carefully considered the comments and tried our best to provide the below point-by-point responses for your comments and questions. Thank you very much for your interest in this manuscript.

Sincerely,

The authors

| I. Referee #2 | |
|---|---|
| **Conceptual issues** | |
| **Comment / Question** | **Our reply** |
| **R2Q1.** 1- The discussion part is too concise and does not depend into the fact that landscape/soil conditions is a major driver in determining land use, and in some extend management within the same land use. As part of the general improvement of the discussion section, I suggest to address this issue in more detail. | Ok, we will improve the discussion in more detail. |
| **R2Q2.** The dataset presents, as it is usual in this kind of large surveys, a large variability. Since the hypothesis to test depends on the ability of | We did not find a strong relationship between tree density and SOC stock in all land uses. For example, in annual cropland $R^2 = 0.05$. |

sampling to represent the soil properties of the plot I wonder how much of this variability for the land uses of density on non-cultivated trees (e.g. annual cropland) comes the variability in soil OC and TN that should be related to the distance to the tree, which according to the sampling method (see line 110) might be a relevant factor. This might be worth discussing in the manuscript, particularly when the higher variability in the regressions (see Figure 6) tend to appear in the land

| | |
|---|---|
| uses with lower non-cultivated tree density. | |
| **R2Q3.** One of the land used (forest) has a very small number of samples (n=2) which might limit the statistical power of the analysis made on this land use. It will be a good idea to include some caveat on this in the result and discussion section, and comment its possible implications. | Indeed, the number of the observed forest was only (02) two. It limited our statistical analysis. Ok, we will explain it more in the results. |
| **R2Q4.** The authors are right, in my view, to claim that their hypothesis is validated for some land uses. In some areas of the world more skilled farmers or | Indeed, we will discuss it more in the manuscript. |

| | |
|---|---|
| shepherds, tend to be also paying more attention to other positive landscape elements, like non-cultivated trees. Perhaps the authors want to consider this as potential underline factor in their discussion, particularly since this been true a more careful management of the grassland and cropland might be taking place. | |
| **R2Q5.** Line 286. "…mainly driven…" I wonder if the authors want to qualify this statement. They have demonstrated that it is a major driver, given some | Yes, we mean "major driver" when we mention "mainly driven" |

| | |
|---|---|
| of the other factors involved perhaps they want to qualify this statement. | |
| **Editing issues** | |
| **R2Q7.** Line 64. There are also many studies in Mediterranean type of climate as compared to tropical regions in Africa. | Ok, we will compare our results to studies in Mediterranean type of climate. |
| **R2Q8.** Line 103. "main crop". Does this mean that the non-cultivated tree has some use, e.g. wood occasionally? Please clarify | Yes, the non-cultivated trees could have multiple uses such as medicinal use, grazing use, firewood collection and nutritional use. According to the native people questioned, for example *Daniella oliveri* Hutch. & Dalz was used for wood collection. *Piliostigma thonningui* (Schum.) Millne-Redhead was used as fodder. *Parkia biglobosa* (Jacq.) Benth. was used as fodder and the pulp and grains was used for human consumption. *Albizia ferruginea* (Guill. & Perr.) Benth. was used as paint in art activities. |
| **R2Q9.** Line 110. Was proximity to the non-cultivate treed considered somehow in the land uses | No, the proximity to the non-cultivated trees to the land uses with low non-cultivated tree density was not considered. We considered the non-cultivated trees in each plot. |

| | |
|---|---|
| with low non-cultivated tree density? | |
| **R2Q10.** Line 120. Indicate also the number of samples, not only the percentage. | The 15 % represent 90 soil samples while the total number of soil samples was 594 soil samples. We will add it in the manuscript. |
| **R2Q11.** Line 123. Some reference (or results of internal test) to validate this assumption? | The pH of these LDSF sampled soils were not measured. However, for other study of my PhD, 38 soil samples were taken in yam fields of the LDSF site at 0-30 cm soil depth. The pH of yam fields soil samples was measured. The results showed that 87% of the soil samples presented a mean pH was 6.05. The pH varied from 5.2 to 6.8. |
| **R2Q12.** Line 125- It is always better to indicate the chemical name not the commercial one. | Calgon is a combination of sodium hexmetaphosphate and sodium carbonate. Ok, we will indicate the chemical name of calgon in the manuscript instead. |
| **R2Q13.** Line 131 add regression y=mx+n in Figures S2, S3 and S4 to allow the reader to see possible biases as compared to the 1:1 line. | Ok, we will add regression $y = mx + n$ in figures S2, S3 and S4. |

| | |
|---|---|
| **R2Q14.** Line 143-145. It is a bit confusing. Did you use bulk density from measurements and coarse fragments (frag) from Hounkpatin el al. (2018) or both from Hounkptin et al. (2018)? Please clarify. Please revise titles of Table S2 to indicate this. | Coarse fragments (frag) was from our measurement while bulk density was from Hounkpatin et al. (2018). As the number of plot of LDSF site was high (160 plots), the auger for soil sampling was the standard manual auger not cylindrical. Thus we estimated the soil volume and we got under-estimated soil volumes that lead to some aberrant results (BD= 2.5 or 2.8 even 3). |
| **R2Q15.** Line 154. I guess that Figure S4 should be Figure S5. Please revise. | Indeed, we will correct it in the manuscript. |
| **R2Q16.** Line 189 Table S3. Indicate what the numbers mean in the Table caption. I have another question. Why not statistical analysis has | Impact of fire, erosion and grazing data were collected by looking in the surrounding plot the visible signs. So, for each land use, the presence of signs was counted and we obtained one data for each land use. Thus, statistical analysis could not be done, we presented those data in percentage in table S3. |

| | |
|---|---|
| been carried out on Fire , Erosion…? | |
| **R2Q17. Line 171.** Where the conditions for normal distribution and variance tested for proper use of ANOVA? Please indicate, or correct of needed. | Yes, the conditions for normal distribution and variance were tested. We used log to normalize the data. We could add the data distribution figure in supplementary material if necessary. |
| **R2Q18.** Line 192. What do you mean by restrictions? That sampling could not be carried out because a rocky horizon was found? Please clarify. In Table 2 indicate n (number of samples) at each depth which can provide a glimpse on how many | Yes, the sampling could not be carried because of the presence of mostly lateritic rocks. In some part of the area lateritic rocks could be observed at soil surface as shown in the below figures. |

times this happened for any given depth.

[Figure]

[Figure]

| | |
|---|---|
| **R2Q19.** Line 231, Section 3.5 An additional Figure showing the cumulate SOC and TN stock with depth should be included (it could go in Supplementary material or in Figures). It can help to provide a complementary view of | Ok, we will add an additional figure showing the cumulate SOC and TN stock with depth including statistical significance of means values. |

| your results and clarify the presentation. It should include an analysis of the statistical significance of differences in mean vales of SOC and TN stock which are not shown now. | |
|---|---|